# Macrophage as a Peripheral Pain Regulator

**DOI:** 10.3390/cells10081881

**Published:** 2021-07-24

**Authors:** Risa Domoto, Fumiko Sekiguchi, Maho Tsubota, Atsufumi Kawabata

**Affiliations:** Laboratory of Pharmacology and Pathophysiology, Faculty of Pharmacy, Kindai University, Higashi-Osaka 577-8502, Japan; risa.domoto@gmail.com (R.D.); fumiko@phar.kindai.ac.jp (F.S.); maho@phar.kindai.ac.jp (M.T.)

**Keywords:** macrophage, neuroimmune crosstalk, neuropathic pain, visceral pain, inflammatory pain, dorsal root ganglion, primary afferent

## Abstract

A neuroimmune crosstalk is involved in somatic and visceral pathological pain including inflammatory and neuropathic components. Apart from microglia essential for spinal and supraspinal pain processing, the interaction of bone marrow-derived infiltrating macrophages and/or tissue-resident macrophages with the primary afferent neurons regulates pain signals in the peripheral tissue. Recent studies have uncovered previously unknown characteristics of tissue-resident macrophages, such as their origins and association with regulation of pain signals. Peripheral nerve macrophages and intestinal resident macrophages, in addition to adult monocyte-derived infiltrating macrophages, secrete a variety of mediators, such as tumor necrosis factor-α, interleukin (IL)-1β, IL-6, high mobility group box 1 and bone morphogenic protein 2 (BMP2), that regulate the excitability of the primary afferents. Neuron-derived mediators including neuropeptides, ATP and macrophage-colony stimulating factor regulate the activity or polarization of diverse macrophages. Thus, macrophages have multitasks in homeostatic conditions and participate in somatic and visceral pathological pain by interacting with neurons.

## 1. Introduction

Somatic or visceral nociceptive pain is an alarm reaction in response to tissue damage (e.g., sprains, bone fracture and burns), abnormal muscle stresses, obstructions, high intraluminal pressure, inflammation, etc. Nociceptors sense the damaged or stimulated parts of the body, and signal the brain mostly via the spinal cord. Persistent or chronic pain that may remain unrelieved for a long time despite the disappearance of the causes, impairs the quality of life in patients, and is often resistant to existing analgesics such as non-steroidal anti-inflammatory drugs and narcotics [1]. Therefore, the development of next-generation analgesics with a new mechanism of action is of paramount clinical importance. Pathological pain is often categorized into neuropathic pain and inflammatory pain. Neuropathic pain is caused by direct or indirect nerve injury in the neural circuitry that mediates pain [2,3,4], while inflammatory pain is driven by inflammatory mediators, e.g., cytokines, released from immune cells [5,6]. Nonetheless, immune cells appear to contribute significantly to not only inflammatory but also neuropathic pain [7,8]. When the primary afferents are injured directly or stimulated indirectly, immune cells including peripheral macrophages or spinal microglia accumulate around the neurons including their cell body and central and peripheral axons through proliferation, infiltration or migration. These immune cells release various pro-inflammatory/pro-nociceptive mediators that act on the nociceptors to drive peripheral sensitization [8]. Recently, the crosstalk between nociceptors and microglia/macrophages in chronic neuronal diseases including pathological pain has been an area of intense and growing research. There are a number of excellent review articles with respect to the role of spinal microglia and astrocytes in pathological pain elsewhere [9,10,11]. Intriguingly, the latest studies characterizing two subsets of tissue-resident macrophages, especially “peripheral nerve macrophages”, originating from fetal and adult monocytes opened new avenue for understanding the role of peripheral macrophages as pain regulators [12,13,14,15]. Here, we focus on the role of peripheral infiltrating and tissue-resident macrophages in the development of somatic and visceral pathological pain including inflammatory and/or neuropathic components.

## 2. Circulating Monocyte-Derived Infiltrating Macrophages and Tissue-Resident Macrophages Including Spinal Microglia and Peripheral Nerve Macrophages

In most tissues, two classes of macrophages, i.e., infiltrating and tissue-resident macrophages, are recognizable in health or disease. The former class of macrophages that infiltrate into tissues in response to injury or infection are derived from circulating monocytes that are produced by hematopoietic stem cells (HSCs) in the post-natal bone marrow. The latter class of macrophages are present in tissues under homeostatic conditions. The bone marrow-derived infiltrating macrophages and tissue-resident macrophages have common and different roles in health and/or disease, and both of them are considered to participate in processing or modulation of pain signals [8,15]. The majority of tissues in the body contain tissue-resident macrophage populations that sense and respond to invading pathogens, environmental stressors or noxious stimuli, and are also essential in tissue development, homeostasis, remodeling and repair [16,17,18,19,20]. Specific tissue-resident macrophage subpopulations, such as microglia in the central nervous system (CNS), Langerhans cells in the skin, and Kupffer cells in the liver, possess a common character of “mononuclear phagocytes”, but reveal extreme heterogeneity that may explain their functional diversity. The heterogeneity is a necessary consequence of tissue-specific and microanatomical niche-specific functions during the development and tissue homeostasis. Apart from the well-described critical role of spinal microglia in the neuropathic and inflammatory pain, there is a growing body of evidence that “sensory neuron-associated macrophages”, a subpopulation of resident peripheral nerve macrophages, participate in the development of peripheral nerve injury-induced neuropathic pain [12,14,15].

It had been believed that tissue-resident macrophages were continuously repopulated by circulating monocytes that arose from progenitors in the adult bone marrow [21]. However, this paradigm was challenged by recent findings that tissue-resident macrophages were seeded during waves of embryonic hematopoiesis and locally self-maintain independently of bone marrow contribution (Figure 1) [16,17]. In mice, embryonic macrophages generate during three distinct waves of hematopoiesis including (i) primitive wave, (ii) transient erythro-myeloid progenitor (EMP) wave, and (iii) definitive wave. In the primitive wave, yolk sac progenitors generate from the “blood islands” in the extra-embryonic yolk sac around embryonic day (E) 7.5 (Figure 1) [17,22]. The primitive yolk sac-derived progenitors give rise to microglia that are a tissue-resident macrophage subpopulation in the CNS throughout adulthood [23]. In the EMP wave, EMPs generate from the yolk sac vascular endothelium around E8.5 (Figure 1). EMPs proliferate in the yolk sac, and start to seed the developing fetal liver through the newly established blood circulation, followed by differentiation into fetal monocytes [24]. The EMP-derived fetal monocytes migrate to most tissues other than the brain, leading to major pools of adult tissue-resident macrophages [17]. In the definitive wave, hematopoiesis is mediated by the definitive HSCs that arise from the intraembryonic hemogenic endothelium of the aorta-gonad-mesonephros (AGM) region (Figure 1). The HSCs rapidly migrate to the fetal liver and seed the fetal bone marrow where these cells will eventually lead to the generation of adult bone marrow HSCs. The definitive HSCs can also generate fetal monocytes that give rise to a minor population of resident macrophages [17]. The primitive yolk sac-derived CNS resident microglia, as well as the fetal monocyte-derived epidermal Langerhans cells, hepatic Kupffer cells and alveolar macrophages proliferate and maintain themselves locally throughout adulthood, whereas the fetal monocyte-derived tissue-resident macrophages in the intestine, dermis, heart and pancreas are replaced slowly by adult circulating monocyte-derived macrophages [18] (Figure 1). There is evidence that the peripheral nerve macrophages in the sciatic nerves lack most of the core signature genes of microglia, and originate primarily from late embryonic precursors, which are slowly replaced by adult circulating monocyte-derived macrophages [12] (Figure 1).

## 3. Polarization of Macrophages

Tissue-resident macrophages play critical roles in the normal tissue homeostasis. On the other hand, in response to tissue injury or inflammatory insults, a number of circulating monocytes recruited from the bone marrow infiltrate the tissue and differentiate into macrophages. The recruited and resident macrophages proliferate and undergo phenotypic and functional changes in response to factors released from immune cells or neurons in the local tissue microenvironment [20,25,26]. Mature macrophages are classically subdivided into two major phenotypes, pro-inflammatory (M1) and anti-inflammatory (M2) macrophages (Figure 2A). M1 macrophages are polarized by bacterial lipopolysaccharide (LPS), interferon-γ (IFN-γ), tumor necrosis factor-α (TNF-α) and granulocyte/macrophage colony-stimulating factor (GM-CSF) (Figure 2A). Functionally, M1 macrophages have potent anti-microbial and anti-tumor activities but cause tissue damage through the release of pro-inflammatory mediators. Such tissue damage and inflammation mediated by M1 macrophages are inhibited later on by M2 macrophages, which are polarized by interleukin (IL)-4, IL-10, IL-13, transforming growth factor-β (TGF-β) and macrophage colony-stimulating factor (M-CSF). M2 macrophages secrete IL-10 and TGF-β to suppress inflammation, have a potent phagocytosis capacity, scavenge debris and apoptotic cells, and promote tissue repair and wound healing. Macrophages are highly plastic cells in the hematopoietic system, because M1 macrophages can switch to M2 phonotype, and vice versa [26,27]. M2 macrophages are further divided into four different subsets, M2a, M2b, M2c and M2d, according to the expression or release of specific proteins and functional characteristics [26]. When the primary afferents and/or adjacent tissues are injured, macrophages accumulate not only around the injured site, but also extensively along their axons and cell body, i.e., dorsal root ganglion (DRG), where M1 macrophages promote excitation of the nociceptive neurons, which might be reduced by M2 macrophages later on [4,28] (Figure 2A).

## 4. Key Mediators Involved in a Macrophage-Nociceptor Crosstalk

### 4.1. Macrophage-Derived Pro-Nociceptive Mediators

A number of mediators released from M1 macrophages, including IL-1β [29,30], IL-6 [31,32], TNF-α [33], nerve growth factor (NGF) [34,35], insulin-like growth factor 1 (IGF-1) [36], prostaglandin E_2_ (PGE_2_) [37,38,39], C-C motif chemokine 2 (CCL2)/monocyte chemoattractant protein 1 (MCP1) and C-X-C motif chemokine ligand (CXCL)1 [7], have been shown to cause neuronal sensitization by stimulating their specific receptors expressed in the nociceptive neurons [4,28,40] (Figure 2B). There is also evidence that macrophages, possibly of the M1 phenotype, once activated, secrete high mobility group box 1 (HMGB1), a nuclear protein [41,42,43,44]. The extracellular HMGB1 causes activation of membrane receptors, such as the receptor for advanced glycation end-product (RAGE) and Toll-like receptor 4 (TLR4), and acceleration of the CXCL12/ C-X-C motif chemokine receptor 4 (CXCR4) signaling, expressed in the primary afferents, contributing to the pathogenesis of inflammatory or neuropathic pain accompanying arthritis, pancreatitis, cystitis and surgically or chemically induced sensory nerve injury [41,42,43,44,45,46,47,48,49,50,51] (Figure 2B). Macrophages also express cystathionine-γ-lyase (CSE), a hydrogen sulfide (H_2_S)-forming enzyme, and CSE upregulation in activated macrophages increases generation of H_2_S and aggravates inflammation [52,53]. Interestingly, H_2_S causes functional upregulation of Ca_v_3.2 T-type Ca^2+^ channels (a member of the low voltage-activated Ca^2+^ channel family) and activates transient receptor potential ankyrin 1 (TRPA1) channels in the sensory neurons, contributing to various intractable pain [54,55,56,57,58,59,60,61,62,63,64]. Taken together, it is likely that macrophage-derived H_2_S mediates pathological pain by targeting Ca_v_3.2 and TRPA1 (Figure 2B). M1 macrophage-derived pro-nociceptive mediators trigger a variety of neuronal cell signals including cytosolic Ca^2+^ mobilization and activation of cyclic AMP/protein kinase A (PKA), phospholipase C (PLC), protein kinase C (PKC), mitogen-activated protein (MAP) kinases (MAPK), nuclear factor-kappa B (NF-κB), Janus kinase (JAK)/signal transducers and activator of transcription (STAT) and Src, leading to neuronal sensitization or excitation [4,5,28,65] (Figure 2B). Activation of PLC, PKC or cyclic AMP/PKA signals enhances the activity of transient receptor potential vanilloid 1 (TRPV1), TRPA1 or Ca_v_3.2 channels, known as pro-nociceptive cation channels, contributing to pathological pain [66,67,68,69] (Figure 2B). On the other hand, M2 macrophages secrete IL-10, which in turn suppresses neuronal excitability and neuropathic pain (Figure 2B), and there is also evidence for anti-nociceptive roles of macrophage-derived opioid peptides [70,71].

### 4.2. Nociceptor-Derived Mediators That Regulate Macrophage Functions

Some mediators released from the sensory neurons are capable of activating macrophages [4,5,8,28,40,65,72]. Substance P (SP) and calcitonin gene-related peptide (CGRP), released from the peripheral nerve terminals of peptidergic neurons, are well-known mediators of neurogenic inflammation. SP stimulates tachykinin NK_1_ receptors expressed in macrophages as well as endothelial cells, leading to increased vascular permeability, vasodilation and edema [28] (Figure 2B). SP-stimulated macrophages secrete pro-inflammatory cytokines, such as IL-1β, TNF-α, macrophage-inflammatory protein-2 and CCL2/MCP1 [73,74]. SP also polarizes macrophages pretreated with IFN-γ to tissue-repairing M2 macrophages [75]. CGRP and its receptors are now well-recognized therapeutic targets for migraine involving neurogenic inflammation [76], whereas there is evidence that CGRP reduces macrophage activity [77] (Figure 2B). ATP is released from neurons most probably through pannexin-1 channels [78,79] (Figure 2B) and activates macrophages via P2X and P2Y receptors [43,80]. Activation of P2X_7_ receptors in macrophages induces IL-1β release via activation of Nod-like receptor protein (NLRP) 3 inflammasome [80,81], which might lead to pyroptosis [81,82]. Activation of P2X_4_ receptors causes release of IL-1β and PGE_2_ from macrophages, participating in pathological pain [83,84]. ATP also triggers macrophage infiltration and HMGB1 release from macrophages through stimulation of P2X_4_ and/or P2X_7_ receptors [43]. Furthermore, CCL2 [85,86] and C-X3-C motif chemokine ligand 1 (CX3CL1) [87], released from the injured sensory neurons, activate macrophages, contributing to neuropathic pain [7] (Figure 2B). Interestingly, non-coding microRNAs (miRs) released from injured neurons contribute to sensory neuron-macrophage communication; e.g., nociceptors are capable of secreting exosomes containing miR-21 that increases M1 macrophages and promotes pain, and also miR-124 that increases M2 macrophages and attenuate persistent inflammatory pain [7,88] (Figure 2B).

## 5. Role of Macrophages in Pathological Pain

### 5.1. Inflammatory Somatic Pain

Circulating monocyte-derived infiltrating macrophages play a major role in inflammatory responses in peripheral tissues. Peripheral HMGB1, which can be released from macrophages under inflammatory conditions, cause long-lasting allodynia/hyperalgesia in rat and mouse hindpaw [44,47,48], and plays a role in the development of inflammatory somatic pain [89]. In rheumatoid arthritis, a variety of inflammatory cells infiltrate into the synovial tissue, and release pro-inflammatory mediators such as TNF-α, IL-1, IL-6, IL-8, IL-12, IL-17, and M-CSF, leading to inflammation followed by nociceptor sensitization [90]. Interestingly, peripheral HMGB1 mediates collagen-antibody-induced arthritis-mediated nociceptor hypersensitivity in male, but not female, mice, and intra-articular injection of HMGB1 causes resident macrophage-dependent mechanical hypersensitivity via activation of TLR4 only in male mice, suggesting sex differences in the HMGB1/TLR4 axis [51]. The many facets of macrophages, including the M1/M2 paradigm, have been described in rheumatoid arthritis [91]. Nociceptor-derived mediators also regulate macrophage activity and inflammatory pain (Figure 2B). SP derived from peptidergic sensory neurons activates pro-inflammatory macrophages via tachykinin NK_1_ receptors followed by inflammatory pain, but is also able to polarize already-activated macrophages to a tissue-repairing M2 macrophage phenotype, followed by attenuation of inflammation and inflammatory pain in mice with collagen II-induced arthritis [92]. Neuroprotectin D1 (NPD1), one of specialized pro-resolving mediators (SPM) derived from essential fatty acid, increases M2 macrophages via activation of GPR37, and reduces inflammatory pain by increasing IL-10 [7]. There is also plenty of clinical and preclinical evidence for the association between macrophages and osteoarthritis [93].

### 5.2. Neuropathic Pain

Neuropathic pain is caused by direct or indirect nerve damage or lesions in the neural circuitry that mediates pain, and involves a neuro-immune crosstalk [8,50,94,95]. Recent evidence indicates that peripheral nerve macrophages, a subpopulation of tissue-resident macrophages, originate primarily from late embryonic precursors and become replaced by bone marrow-derived macrophages over time (see Figure 1), and that monocyte-derived infiltrating macrophages following nerve injury can engraft in the pool of resident peripheral nerve macrophages, contributing to the development of neuropathic pain [12,15]. A number of mediators, such as ATP, SP, CCL2 and CX3CL1 derived from neurons, promote infiltration of adult monocyte-derived macrophages and/or proliferation of resident macrophages after nerve injury, leading to macrophage accumulation around nociceptors including the DRG [7,28,78,79,87,94] and then afterwards a variety of pro-nociceptive mediators derived from macrophages induce neuronal sensitization or excitation (Figure 2B).

There is plenty of evidence that monocyte-derived infiltrating macrophages and tissue-resident peripheral nerve macrophages are involved in neuropathic pain following surgical injury of the sciatic nerves or spinal nerves [15,44,50,94]. We have shown that L5 spinal nerve injury causes macrophage accumulation in L4 DRG, in which macrophage-derived HMGB1 induces RAGE-dependent transcriptional upregulation of Ca_v_3.2 T-type Ca^2+^ channels in nociceptors, leading to neuropathic pain [50]. Intriguingly, sciatic nerve injury causes upregulation of miR-21 in the DRG, and subsequently release of miR-21-containing exosomes from the cell body of nociceptor neurons, which are readily phagocytosed by macrophages, leading to neuropathic pain through promotion of M1 macrophages and inhibition of M2 macrophages [88]. On the other hand, it is noteworthy that macrophage-derived miR-155 inhibits axonal regeneration after spinal cord injury [96].

Diabetic peripheral neuropathy is a common complication associated with diabetes. In streptozotocin (STZ)-induced type 1 diabetic rats, adult monocyte-derived macrophages accumulate in the sciatic nerves, contributing to painful diabetic neuropathy [97,98]. Neuroinflammation driven by circulating monocyte-derived infiltrating macrophages and resident peripheral nerve macrophages is also considered to mediate complications, including painful neuropathy associated with type 2 diabetes [99,100].

Chemotherapy-induced peripheral neuropathy (CIPN) frequently develops in cancer patients treated with anticancer agents, such as paclitaxel, oxaliplatin, vincristine, bortezomib, etc. [101] The impact of macrophages on the pathogenesis of CIPN appears to differ depending on the chemotherapeutic agents [42,44]. The best known chemotherapeutic agent that causes CIPN through macrophage accumulation in the DRG and sciatic nerves in mice and rats is paclitaxel, a microtubule stabilizer [41,86,87,102]. The accumulating macrophages appear to contain circulating adult monocyte-derived infiltrating macrophages and also fetal and adult monocyte-derived resident peripheral nerve macrophages that might proliferate after paclitaxel treatment [15] (Figure 3). It has long been known that paclitaxel directly stimulates macrophages [103,104]. We have shown that paclitaxel directly stimulates macrophages, which in turn secrete HMGB1 through upregulation of histone acetyltransferases that trigger cytoplasmic translocation of nuclear HMGB1, via activation of the reactive oxygen species (ROS)/p38MAPK/NF-κB pathway [41]. The CIPN development following paclitaxel treatment can be prevented by an anti-HMGB1-neutralizing antibody, macrophage/microglia inhibitor (minocycline), macrophage depletor (liposomal clodronate), NF-κB inhibitor, antioxidant, or soluble thrombomodulin that accelerates HMGB1 degradation by thrombin [41,42,44]. The extracellular HMGB1 derived from macrophages in response to paclitaxel is considered to enhance neuronal excitability by activating RAGE and enhancing CXCL12/CXCR4 signaling, resulting in CIPN [41] (Figure 3), while peripheral TLR4, unlike spinal TLR4, does not appear to play a role in CIPN following paclitaxel treatment [41,105]. Paclitaxel also directly induces release of TNF-α and CXCL1 from macrophages, which in turn enhances neuronal excitability essential for CIPN in mice [106]. There is evidence that neuron-derived CCL2 and CX3CL1 induce macrophage accumulation into DRG following paclitaxel, leading to CIPN [86,87] (Figure 3). Paclitaxel causes expression of NLRPs in macrophages, leading to release of IL-1β that sensitizes sensory neurons responsible for CIPN in rats [107] (Figure 3). The role of macrophages in CIPN caused by chemotherapeutics other than paclitaxel is controversial. We have demonstrated that oxaliplatin treatment in mice does not cause notable macrophage accumulation in the DRG or sciatic nerves, and that HMGB1 derived from non-macrophage cells participates in CIPN following oxaliplatin treatment in mice [42,44,108], although a study has suggested only a limited role of macrophages in CIPN caused by oxaliplatin in mice [109].

### 5.3. Visceral Pain

Neuron-glia-immune interaction in the spinal cord, DRG and enteric nervous system is now considered to play a role in the development of visceral hypersensitivity, which is associated with visceral pain accompanying inflammatory bowel disease (IBD), irritable bowel syndrome (IBS), and interstitial cystitis (IC)/bladder pain syndrome (BPS) [110]. Interestingly, injury to the primary sensory nerve innervating the colon during colonic inflammation causes macrophage clustering around the cell body of the sensory neurons of the DRG. In DRG, M-CSF, also known as CSF1, derived from the cell body of the injured sensory nerves, stimulates the proliferation and activation of resident peripheral nerve macrophages [94], and the activated macrophages in turn enhance the neuronal excitability and activate satellite glial cells (SGCs) by producing pro-inflammatory/pro-nociceptive mediators including TNF-α, IL-1β, and IL-6, resulting in colonic pain/hypersensitivity [110,111] (Figure 4). There is evidence that macrophage-derived TNF-α in DRG reduces the expression of Kir4.1, an inwardly rectifying K^+^ channel, in SGCs, contributing to SGC activation [112] (Figure 4). Thus, macrophages and SGCs orchestrate to enhance and prolong colonic pain. The macrophage-dependent SGC activation also plays a role in cross-organ sensitization, e.g., co-occurring bladder and colonic dysfunction/hypersensitivity. Gap junctions between adjacent SGCs mediate transmission of neuronal excitability from the injured primary sensory neurons following colitis to the non-injured primary sensory neurons innervating the bladder [110,113] (Figure 4). Thus, gut-bladder cross sensitization appears to involve coupling of SGCs by gap junctions in DRG, in addition to the activation of microglia and astrocytes in the spinal cord. In the intestinal enteric nervous system, intrinsic enteric neurons, extrinsic primary afferent neurons, enteric glial cells (EGCs), and muscularis macrophages interact with each other, thereby regulating colonic sensitivity and motility (Figure 4). Muscularis macrophages, a subpopulation of intestinal tissue-resident macrophages, are largely derived from bone marrow progenitors, but contain a subpopulation of long-lived and self-maintaining resident macrophages [114,115]. Colonic tissue injury causes excitation of extrinsic primary afferents and the release of M-CSF from intrinsic enteric neurons, which in turn polarizes muscularis macrophages to pro-inflammatory/pro-nociceptive macrophages, resulting in neuronal sensitization and colonic hypersensitivity [116] (Figure 4). M-CSF, essential for M1 polarization of muscularis macrophages, is also released from EGCs in response to colonic tissue injury [117], and bone morphogenic protein 2 (BMP2) derived from muscularis macrophages acts on intrinsic enteric neurons, thereby regulating smooth muscle contractility and peristalsis [115,116] (Figure 4).

IC/BPS with and without Hunner lesions is characterized by pelvic/suprapubic pain, bladder discomfort or pressure and increased frequency of urination, and associated with peripheral and central nerve sensitization [118,119]. Migration of bone marrow-derived macrophages to the bladder tissue has been reported in different animal models for IC/BPS [43,120,121]. We have shown that, in the cyclophosphamide (CPA)-induced cystitis/bladder pain model in mice, CSE, an H_2_S-generating enzyme, and Ca_v_3.2 T-type Ca^2+^ channels, which are functionally accelerated by H_2_S, are upregulated in the urothelial cells and primary afferents, respectively, contributing to the induction of bladder pain [43,62,122,123]. On the other hand, we have demonstrated the involvement of endogenous HMGB1 and its receptor RAGE in the development of CPA-induced bladder pain [46]. Our latest study has provided evidence that the urothelium, once stimulated with acrolein, a hepatic metabolite of CPA, releases ATP, which in turn causes migration of circulating monocyte-derived macrophages to the bladder tissue via activation of P2X_4_ and P2X_7_ receptors, followed by P2X_7_-dependent release of HMGB1 from the activated macrophages, and that the extracellular HMGB1 induces RAGE-dependent upregulation of CSE in the urothelium, leading to activation of the H_2_S/Ca_v_3.2 cascade followed by bladder pain [43] (Figure 5). It is noteworthy that the HMGB1/RAGE signal can cause the early growth response 1 (EGR-1)-dependent overexpression of Ca_v_3.2 in the primary afferent [50,122,123]. Macrophage-derived HMGB1 is also involved in pancreatic pain accompanying acute pancreatitis caused by repeated systemic administration of cerulein in mice [49]. In this pancreatitis-related pain model, adult monocyte-derived macrophages migrate into the pancreatic tissue during the early stage of tissue damage or inflammation, and secrete HMGB1, which in turn activates RAGE and accelerates CXCL12/CXCR4 signaling, resulting in pancreatic pain (Figure 6). Endometriosis-related hypersensitivity of sensory nerves also involves the interaction between macrophages and nerve fibers [36,124].

## 6. Therapeutic Avenue for Pathological Pain by Pharmacological Intervention of the Macrophage–Neuron Crosstalk

Accumulating evidence for the macrophage–neuron crosstalk involved in pain regulation may provide a novel therapeutic avenue for patients with pathological pain. Among cytokines that induce polarization from M0 to M2 macrophages (see Figure 2A), IL-4, when administered locally, induces re-polarization from M1 to M2 macrophages, leading to sustained suppression of neuropathic pain in mice [125]. Similarly, CU-CPT22, a TLR2 antagonist [126], parthenolide, a bioactive compound of Chrysanthemum parthenium L. [127], and erythropoietin [128] may also reduce intractable pain, because of their capability of inducing re-polarization from M1 to M2 macrophages. There is also clinical evidence that tocilizumab, an anti-IL6 receptor monoclonal antibody, infliximab, adalimumab and certolizumab, anti-TNF-α monoclonal antibodies, and canakinumab, an anti-IL-1β monoclonal antibody, reduce neuropathic pain in humans [129]. Macrophage-derived HMGB1, a pro-nociceptive molecule of damage-associated molecular patterns (DAMPs), can be degraded by the endothelial thrombomodulin/thrombin system, an effect mimicked by recombinant human soluble thrombomodulin, known as thrombomodulin alfa (TMα; ART-1213, Recomodulin^®^), which has been approved as a therapeutic medicine for treatment of disseminated intravascular coagulation (DIC) in Japan [42,44]. We have shown that TMα potently suppresses the development of inflammatory pain [47], CIPN [41,45,108] and visceral pain [43,46,49] by promoting thrombin-dependent degradation of HMGB1 [44]. The clinical effectiveness of TMα on CIPN has been demonstrated in colorectal cancer patients undergoing oxaliplatin-based chemotherapy [130].

## 7. Conclusions

Tissue-resident macrophages, as well as bone marrow-derived infiltrating macrophages, interact with the primary sensory neurons in the peripheral tissues and DRG, and regulate not only inflammatory responses but also pain signals, as do microglial cells in the CNS. Recent studies have uncovered previously unknown characteristics of tissue-resident macrophages, such as their origins and association with regulation of pain signals. Particularly, peripheral nerve macrophages and intestinal resident macrophages including muscularis macrophages are considered to have multitasks in homeostatic conditions and participate in somatic and visceral pathological pain containing neuropathic and inflammatory components by interacting with neurons [12,14,15,110,115,116,117]. A number of substances derived from the primary afferent neurons and from macrophages have been identified as mediators for the neuroimmune crosstalk. Thus, recent increasing evidence for the macrophage–neuron interaction, which is drug-targetable for treatment of pathological pain, has shed light on the role of macrophages as peripheral pain regulators.

## Figures and Tables

**Figure 1 cells-10-01881-f001:**
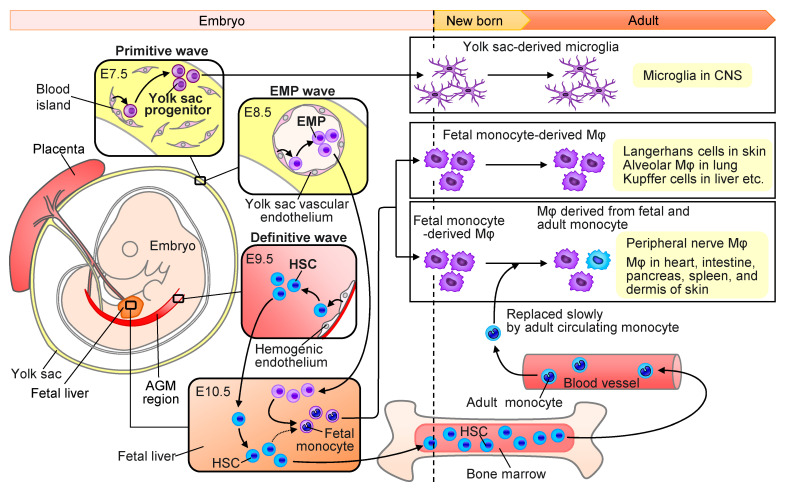
Tissue-resident macrophage (Mφ) generation through three main waves of hematopoiesis during development in mice. “Primitive wave” arises from the blood islands of the extra-embryonic yolk sac at E7.5, and produces primitive yolk sac progenitors, leading to adult microglia in the central nervous system (CNS). The “Erythro-myeloid progenitor (EMP)” wave arises from the hemogenic endothelium formed at E8.5 in the yolk sac and produces EMPs, which migrate into the fetal liver, expand and differentiate into fetal monocytes, leading to adult tissue-resident macrophages in every tissue other than CNS. The “Definitive wave” arises from the intraembryonic hemogenic endothelium of the aorta-gonad-mesonephros (AGM) region, and produces fetal hematopoietic stem cells (HSCs) at E9.5, which colonize the fetal liver and seed the fetal bone marrow, leading to adult HSCs. Some of fetal HSCs might differentiate to fetal monocytes in the fetal liver. The peripheral tissue-resident macrophages proliferate and self-maintain. The fetal monocyte-derived epidermal Langerhans cells, hepatic Kupffer cells, and alveolar macrophages locally self-maintain throughout adulthood, while other tissue-resident macrophages derived from fetal monocytes are replaced slowly by adult-circulating monocyte-derived macrophages.

**Figure 2 cells-10-01881-f002:**
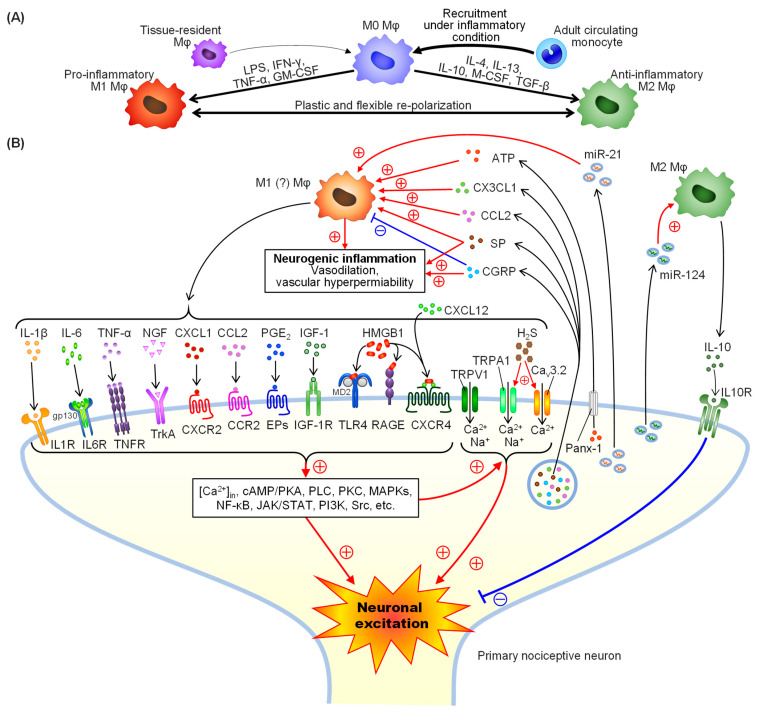
Macrophage (Mφ) polarization and messengers essential for a crosstalk between macrophages and primary afferent neuros. (**A**) M0 macrophages derived from tissue-resident macrophages and adult-circulating monocyte-derived infiltrating macrophages are polarized to pro-inflammatory M1 phenotype by lipopolysaccharide (LPS), interferon-γ (IFN-γ), tumor necrosis factor-α (TNF-α) and granulocyte-macrophage colony-stimulating factor (GM-CSF), and to anti-inflammatory M2 phenotype by interleukin (IL)-4, IL-10, IL-13, transforming growth factor-β (TGF-β) and macrophage colony-stimulating factor (M-CSF). (**B**) A number of messengers derived from M1 macrophages and primary sensory neurons participate in the macrophage–neuron interaction. M1 macrophages release IL-1β, IL-6, TNF-α, nerve growth factor (NGF), C-X-C motif chemokine ligand 1 (CXCL1), C-C motif chemokine ligand 2 (CCL2), prostaglandin E_2_ (PGE_2_) and insulin-like growth factor 1 (IGF-1) activate their receptors, IL1R, IL6R, TNFR, TrkA, CXC chemokine receptor (CXCR) 2, C-C motif chemokine receptor (CCR) 2, E prostanoid receptors (EPs) and IGF-1R, respectively, expressed on the primary afferents. A partially oxidized form of high mobility group box 1 (HMGB1) stimulates Toll-like receptor 4 (TLR4), and a fully reduced form of HMGB1 activates receptor of advanced glycation end-products (RAGE), and accelerates the activation of CXCR4 by CXCL12. Macrophage-derived H_2_S accelerates the activity of Ca_v_3.2 T-type Ca^2+^ channels and activates transient receptor potential (TRP) ankyrin 1 (TRPA1) channels. Neuronal cell signals in response to macrophage-derived mediators include increase in intracellular Ca^2+^ levels ([Ca^2+^]_in_) and activation of the cyclic AMP (cAMP)/protein kinase A (PKA) pathway, phospholipase C (PLC), PKC, MAP kinases (MAPKs), nuclear factor-κB (NF-κB), the Janus kinase (JAK)/signal transducer and activator of transcription (STAT) pathway, phosphatidylinositol-3 kinase (PI3K), Src, etc., which may in turn functionally upregulate TRPA1, TRPV1 and Ca_v_3.2 T-type Ca^2+^ channels. The excitability of the primary afferents can be suppressed by M2 macrophage-derived IL-10. On the other hand, neuron-derived substance P (SP), C-X3-C chemokine ligand-1 (CX3CL1) and CCL2 stimulates pro-inflammatory macrophages. Calcitonin gene-related peptide (CGRP) released from the nociceptors rather suppresses macrophages, but participates in neurogenic inflammation. ATP is released from the sensory neurons through pannexin-1 (Panx-1) and stimulates pro-inflammatory macrophages. Micro RNA (miR)-21 and miR-124 released from injured neurons polarize macrophages to M1 and M2 phenotypes, respectively.

**Figure 3 cells-10-01881-f003:**
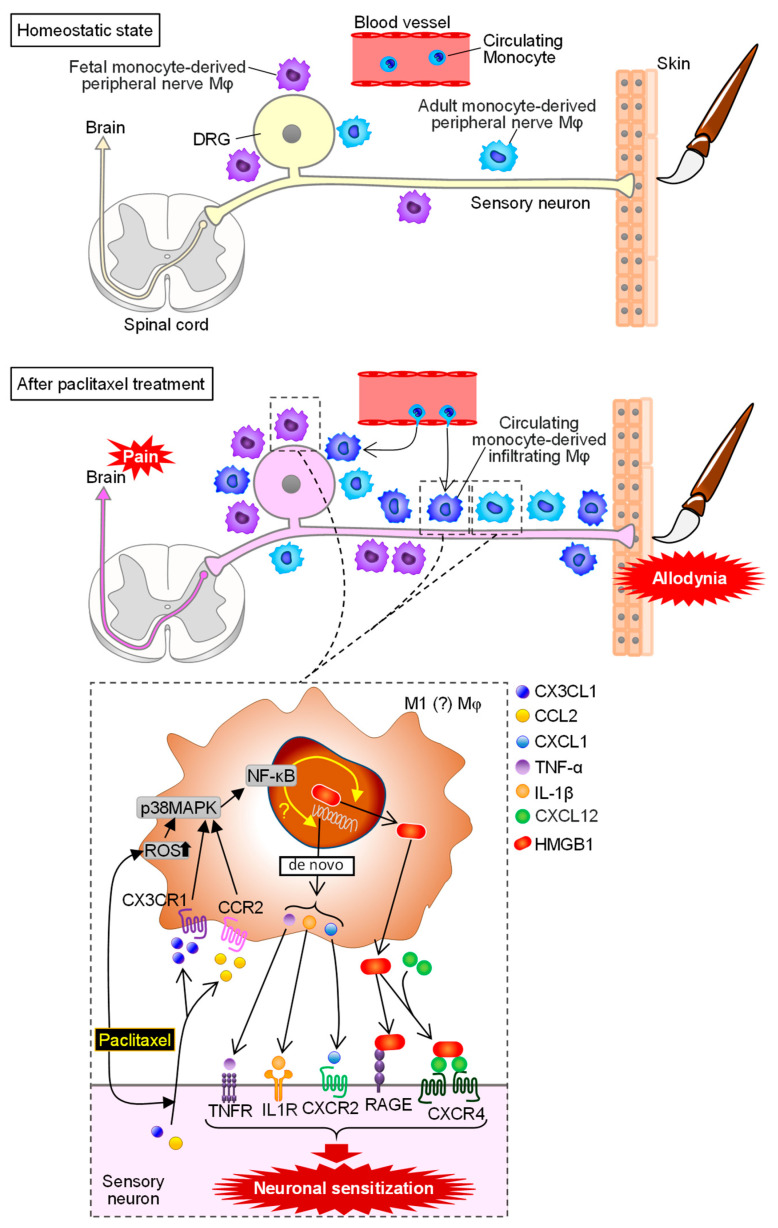
Role of macrophages in chemotherapy-induced peripheral neuropathy (CIPN) caused by paclitaxel treatment. Peripheral nerve macrophages (Mφ), derived from fetal and adult monocytes, self-maintain around the primary afferent. After paclitaxel treatment, macrophages accumulate in the sciatic nerve and DRG through proliferation of peripheral nerve macrophages and infiltration of circulating adult monocyte-derived macrophages. Paclitaxel directly stimulates M1 macrophages and causes release of nuclear HMGB1 through the reactive oxygen species (ROS)/p38MAPK/NF-κB pathway, which in turn induces neuronal sensitization via activation of a receptor for advanced glycation end-product (RAGE) and acceleration of C-X-C motif chemokine ligand (CXCL) 12/C-X-C motif chemokine receptor (CXCR) 4 signals. Paclitaxel also directly increases de novo expression and release of tumor necrosis factor (TNF)-α, interleukin (IL)-1β and CXCL1 in macrophages possibly via NF-κB signals, which in turn enhance neuronal excitability via their receptors, TNFR, IL-1R and CXCR2. On the other hand, paclitaxel causes production of C-C motif chemokine ligand (CCL) 2 and C-X3-C chemokine ligand (CX3C) 1 that activate macrophages via C-C motif chemokine receptor (CCR) 2 and C-X3-C chemokine receptor (CX3CR) 1, respectively.

**Figure 4 cells-10-01881-f004:**
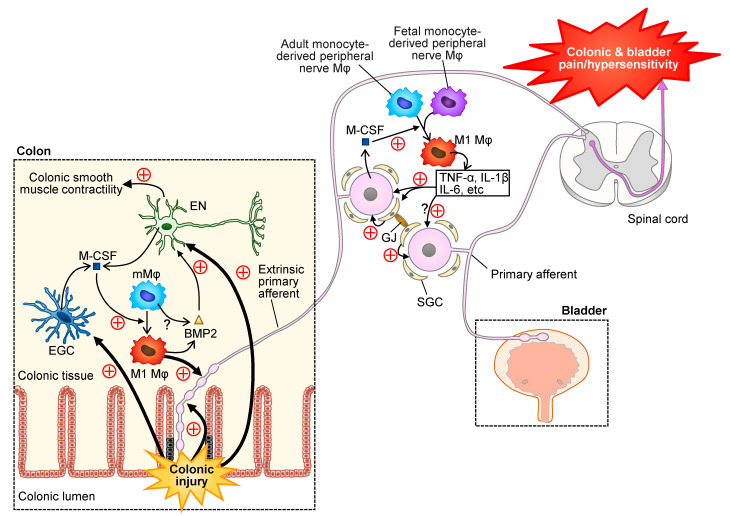
Neuron-glia-macrophage (Mφ) interaction in the DRG and enteric nervous system participates in colonic pain/hypersensitivity followed by bladder pain/hypersensitivity. Colonic injury causes stimulation of the extrinsic primary afferents, enteric nerves (ENs) and enteric glial cells (EGCs). Macrophage colony-stimulating factor (M-CSF) derived from the activated ENs and EGCs stimulates the polarization of muscularis macrophages (mMφ) to pro-inflammatory M1 macrophages that enhance the excitability of the primary afferents. Macrophage-derived bone morphogenic protein 2 (BMP2) stimulates ENs that regulate colonic muscle contractility. In DRG, the injured primary afferents following colonic inflammation release M-CSF, which stimulates and polarizes peripheral nerve macrophages derived from fetal and adult monocytes to M1 fenotype. The activated macrophages release pro-nociceptive cytokines including TNF-α, IL-1β and IL-6, thereby stimulating the primary afferent neurons and satellite glial cells (SGCs). Gap junctions between adjacent SGCs mediate transmission of neuronal excitability from the injured primary sensory neurons following colitis to the non-injured primary sensory neurons innervating the bladder, leading to cross-organ sensitization, i.e., co-occurring colonic and bladder pain/hypersensitivity.

**Figure 5 cells-10-01881-f005:**
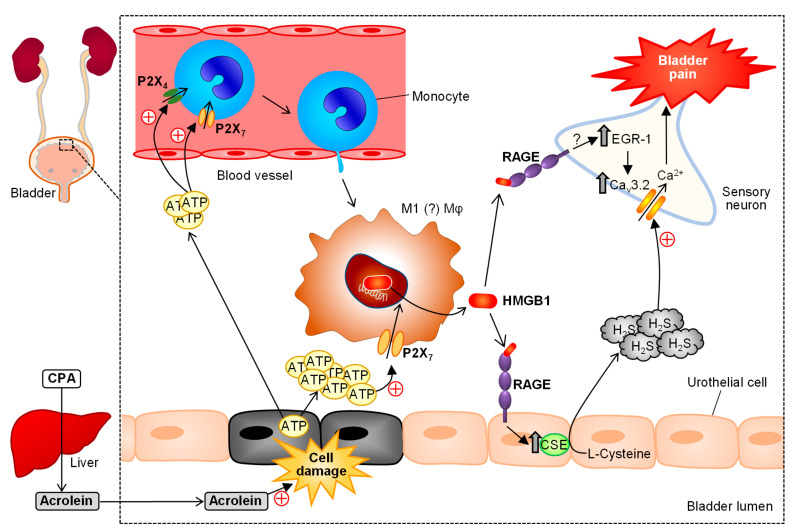
Role of macrophage-derived HMGB1 in the development of cystitis-related bladder pain in mice treated with cyclophosphamide (CPA). Acrolein, a toxic metabolite of CPA, accumulates in the bladder lumen and injures urothelial cells, leading to ATP release. The extracellular ATP causes infiltration of circulating monocytes into the bladder tissue via P2X_4_ and P2X_7_ receptors, and release of nuclear HMGB1 via P2X_7_ receptors. The extracellular HMGB1 induces RAGE-dependent overexpression of cystathionine-γ-lyase (CSE), an H_2_S-generating enzyme, in the urothelium, and the increased H_2_S enhances neuronal Ca_v_3.2 channel activity, leading to bladder pain. The HMGB1/RAGE pathway may also be responsible for the EGR-1-dependent transcriptional upregulation of Ca_v_3.2 in the sensory neuron following CPA treatment.

**Figure 6 cells-10-01881-f006:**
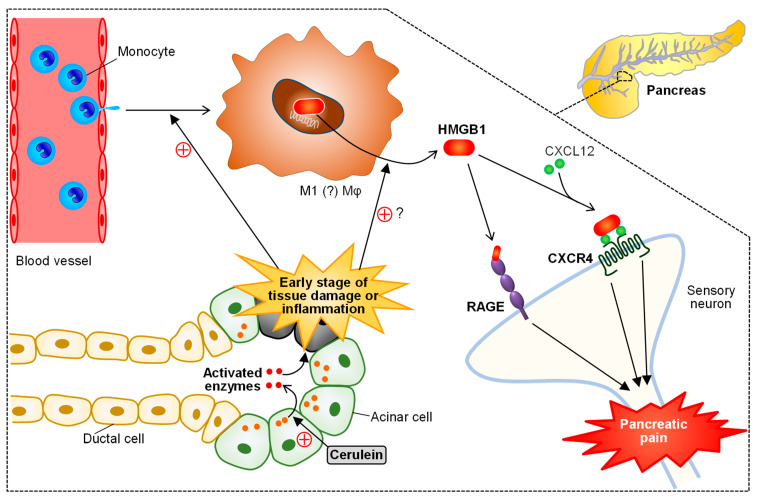
Role of macrophage-derived HMGB1 in the development of pancreatitis-related visceral pain in mice treated with cerulein. In the early stage of cerulein-induced acute pancreatitis, circulating monocyte-derived infiltrating macrophages release HMGB1, which in turn activates the receptor for advanced glycation end-products (RAGE) and accelerates C-X-C motif chemokine ligand (CXCL) 12/C-X-C motif chemokine receptor (CXCR) 4 signals, leading to pancreatic pain.

## Data Availability

Not applicable.

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
