# Peer review of "Macrophage as a Peripheral Pain Regulator"

_cells, 2021, doi:10.3390/cells10081881_

Round 1
Reviewer 1 Report
Macrophage as a peripheral pain regulator_cells-1299697_peer-review
The work by Domoto and colleagues represents an excellent overview of the role of macrophages in peripheral tissues as mediators of the neuroimmune crosstalk that is involved in somatic and visceral pathological pain. The manuscript is well organized and articulated. The narrative flow is logical and easy to follow. The text is enriched by diagrams rendering a clear visualization of the structures, cell populations, and molecular mechanisms involved.
The only weak aspect of the review is the omission of a comprehensive elaboration on how the current knowledge on the mechanisms that regulate macrophage biology in the context of somatic and visceral pathological pain have been or can be translated into effective/promising therapeutic avenues. A more extensive discussion of this topic is warranted to complement the current content and further improve the quality of this interesting review.
Other minor points to consider are reported below:
Use of acronyms and abbreviations – Please, make sure that when acronyms and abbreviations are used for the first time, they are spelled out and followed by the actual acronym and abbreviation in parenthesis. This has not been consistently done throughout the manuscript (see for example NGF).
Abstract, line 8 – Consider replacing “containing” with “including”.
Abstract, line 15 – Replace “secret” with “secrete”.
Abstract, line 18 – Avoid using “etc.” as it is not clear to the reader what comes next in the list.
Introduction, lines 32-33 – Consider replacing “is in urgent need” with “is of paramount clinical importance”.
Introduction, line 38 – Consider rephrasing as follows: “…primary afferents are injured directly or stimulated indirectly, immune cells…”.
Introduction, lines 40-41 – “…throughout the sensory neurons including the peripheral and central axons and the cell body…” does this sentence refer to the distinct structure of the sensory pseudounipolar cells of the dorsal root ganglia and other sensory ganglia of the cranial nerves? If this is the case, please specify.
Introduction, lines 43 – Consider replacing “a crosstalk” with “the crosstalk”.
Introduction, lines 52 – Consider replacing “containing” with “including”.
Circulating monocyte-derived…, lines 62-63 – “…respond to invading pathogens and challenging surroundings…” it is not entirely clear what the authors mean with “challenging surroundings”. Consider rephrasing using another designation. Consider replacing with “environmental stressors”, “noxious stimuli”,or something similar.
Circulating monocyte-derived…, line 70 – Consider replacing “adulthood” with “tissue homeostasis”.
Circulating monocyte-derived…, lines 71-74 – Consider rephrasing as follows: “growing body of evidence that “sensory neuron-associated macrophages”, a subpopulation of resident peripheral nerve macrophages, participate in the development of peripheral nerve injury-induced neuropathic pain”.
Circulating monocyte-derived…, lines 71-74 – Consider rephrasing as follows: “three distinct waves of hematopoiesis including (i) primitive wave, (ii) transient erythro-myeloid progenitor (EMP) wave, and (iii) definitive wave”.
Circulating monocyte-derived…, lines 95-99 – Please, specify whether resident microglia is maintained locally over time or replaced by adult circulating monocytes.
Key mediators involved in a macrophage-nociceptor crosstalk, lines 173-174 – Itchiness is not likely considered a typical manifestation of chronic pathologic pain. Please, specify.
Key mediators involved in a macrophage-nociceptor crosstalk, line 216 – Is there any evidence that the imflammasome activation in this case leads to pyroptosis?
Role of macrophages in pathological pain, lines 332-333 – Consider rephrasing as follows: “clustering around the cell body of the sensory neurons of the DRG”.
Fig. 4 – The diagram of the colonic tissue includes villus-like structure on the mucosal versant. Colonic mucosa doesn’t feature villi. Therefore, it would be more appropriate is the present diagram with villi is replaced by a diagram featuring a mucosal surface without villi.
Author Response
Responses to Reviewer #1
We thank Reviewer #1 for the positive comments.
R1-1)The only weak aspect of the review is the omission of a comprehensive elaboration on how the current knowledge on the mechanisms that regulate macrophage biology in the context of somatic and visceral pathological pain have been or can be translated into effective/promising therapeutic avenues. A more extensive discussion of this topic is warranted to complement the current content and further improve the quality of this interesting review.
We thank the reviewer for suggesting important points that will be useful to improve our manuscript. Considering the reviewer’s opinion, we have incorporated a paragraph “6. Therapeutic avenue for pathological pain by pharmacological intervention of the macrophage-neuron crosstalk” with additional references into the revised manuscript, as follows: Accumulating evidence for the macrophage-neuron crosstalk involved in pain regulation may provide a novel therapeutic avenue for patients with pathological pain. Among cytokines that induce polarization from M0 to M2 macrophages (see Figure 2A), IL-4, when administered locally, induces re-polarization from M1 to M2 macrophages, leading to sustained suppression of neuropathic pain in mice. Similarly, CU-CPT22, a TLR2 antagonist, parthenolide, a bioactive compound of Chrysanthemum parthenium L., and erythropoietin may also reduce intractable pain, because of their capability of inducing re-polarization from M1 to M2 macrophages. There is also clinical evidence that tocilizumab, an anti-IL6 receptor monoclonal antibody, infliximab, adalimumab and certolizumab, anti-TNF-α monoclonal antibodies, and canalinumab, an anti-IL-1 monoclonal antibody, appear to reduce neuropathic pain in humans. Macrophage-derived HMGB1, a pro-nociceptive molecule of damage associated molecular patterns (DAMPs), can be degraded by the endothelial thrombomodulin/thrombin system, an effect mimicked by recombinant human soluble thrombomodulin, known as thrombomodulin alfa (TMα; ART-1213, Recomodulin®), which has been approved as a therapeutic medicine for treatment of disseminated intravascular coagulation (DIC) in Japan. We have shown that TMα potently suppresses the development of inflammatory pain {Tanaka, 2013 #235}, CIPN and visceral pain by promoting thrombin-dependent degradation of HMGB1. The clinical effectiveness of TMα on CIPN has been demonstrated in colorectal cancer patients undergoing oxaliplatin-based chemotherapy. Please find L418-L441 in the revised manuscript.
R1-2) Use of acronyms and abbreviations – Please, make sure that when acronyms and abbreviations are used for the first time, they are spelled out and followed by the actual acronym and abbreviation in parenthesis. This has not been consistently done throughout the manuscript (see for example NGF).
According to the reviewer, I have confirmed acronyms and abbreviations in the text, and spelled out them, if not. Please find L66, L108, L146, L147, L166-L169, L203, L256, L307 in the revised manuscript.
R1-3) Abstract, line 8 – Consider replacing “containing” with “including”.
Yes. It has been retyped according to the reviewer’s suggestion. Please find L8 in the revised manuscript.
R1-4) Abstract, line 15 – Replace “secret” with “secrete”.
Yes. It has been retyped in Abstract and the text, according to the reviewer’s suggestion. Please find L15 and LL151 in the revised manuscript.
R1-5) Abstract, line 18 – Avoid using “etc.” as it is not clear to the reader what comes next in the list.
Yes. It has been altered according to the reviewer’s suggestion. Please find L17-L18 in the revised manuscript.
R1-6) Introduction, lines 32-33 – Consider replacing “is in urgent need” with “is of paramount clinical importance”.
Yes. It has been retyped according to the reviewer’s suggestion. Please find L32-L33 in the revised manuscript.
R1-7) Introduction, line 38 – Consider rephrasing as follows: “…primary afferents are injured directly or stimulated indirectly, immune cells…”.
Yes. It has been retyped according to the reviewer’s suggestion. Please find L39 in the revised manuscript.
R1-8) Introduction, lines 40-41 – “…throughout the sensory neurons including the peripheral and central axons and the cell body…” does this sentence refer to the distinct structure of the sensory pseudounipolar cells of the dorsal root ganglia and other sensory ganglia of the cranial nerves? If this is the case, please specify.
The original sentence might not be clear. Considering the reviewer’s comment, we have just simply described as follows: “-----, immune cells including peripheral macrophages or spinal microglia accumulate around the neurons including their cell body and central and peripheral axons through proliferation, infiltration or migration”. Please find L39-L41 in the revised manuscript.
R1-9) Introduction, lines 43 – Consider replacing “a crosstalk” with “the crosstalk”.
Yes. It has been retyped according to the reviewer’s suggestion. Please find L43 in the revised manuscript.
R1-10) Introduction, lines 52 – Consider replacing “containing” with “including”.
Yes. It has been retyped according to the reviewer’s suggestion. Please find L52 in the revised manuscript.
R1-11) Circulating monocyte-derived…, lines 62-63 – “…respond to invading pathogens and challenging surroundings…” it is not entirely clear what the authors mean with “challenging surroundings”. Consider rephrasing using another designation. Consider replacing with “environmental stressors”, “noxious stimuli”,or something similar.
We thank the reviewer for providing useful opinions. We have retyped the sentence, according to the reviewer’s opinions. Please find L64 in the revised manuscript.
R1-12) Circulating monocyte-derived…, line 70 – Consider replacing “adulthood” with “tissue homeostasis”.
Yes. It has been retyped according to the reviewer’s suggestion. Please find L70 in the revised manuscript.
R1-13) Circulating monocyte-derived…, lines 71-74 – Consider rephrasing as follows: “growing body of evidence that “sensory neuron-associated macrophages”, a subpopulation of resident peripheral nerve macrophages, participate in the development of peripheral nerve injury-induced neuropathic pain”.
Yes. It has been retyped according to the reviewer’s suggestion. Please find L72-L74 in the revised manuscript.
R1-14) Circulating monocyte-derived…, lines 71-74 – Consider rephrasing as follows: “three distinct waves of hematopoiesis including (i) primitive wave, (ii) transient erythro-myeloid progenitor (EMP) wave, and (iii) definitive wave”.
Yes. It has been retyped according to the reviewer’s suggestion. Please find L80-L81 in the revised manuscript.
R1-15) Circulating monocyte-derived…, lines 95-99 – Please, specify whether resident microglia is maintained locally over time or replaced by adult circulating monocytes.
Considering the reviewer’s comments, we have modified the text as follows: The primitive yolk sac-derived CNS resident microglia as well as the fetal monocyte-derived epidermal Langerhans cells, hepatic Kupffer cells, and alveolar macrophages proliferate and maintain themselves locally throughout adulthood, whereas ------. Please find L95-98 in the revised manuscript.
R1-16) Key mediators involved in a macrophage-nociceptor crosstalk, lines 173-174 – Itchiness is not likely considered a typical manifestation of chronic pathologic pain. Please, specify.
According to the reviewer’s opinions, the description about itchiness and IL-31 has been deleted. Therefore, IL-31 in Figure 2 has also been deleted. Please find L173-L175, L196 and new Figure 2B in the revised manuscript.
R1-17) Key mediators involved in a macrophage-nociceptor crosstalk, line 216 – Is there any evidence that the imflammasome activation in this case leads to pyroptosis?
Yes, there is evidence that activation of P2X7 receptors leads to pyroptosis. Considering the reviewer’s comments, we have modified the text and added references, as follows: “Activation of P2X7 receptors ---, which might lead to pyroptosis”. Please find L216 in the revised manuscript.
R1-18) Role of macrophages in pathological pain, lines 332-333 – Consider rephrasing as follows: “clustering around the cell body of the sensory neurons of the DRG”.
Yes. It has been retyped according to the reviewer’s suggestion. Please find L333-L334 in the revised manuscript.
R1-19) Fig. 4 – The diagram of the colonic tissue includes villus-like structure on the mucosal versant. Colonic mucosa doesn’t feature villi. Therefore, it would be more appropriate is the present diagram with villi is replaced by a diagram featuring a mucosal surface without villi.
According to the reviewer’s suggestion, the diagram in Figure 4 has been replaced by one without villi. Please find new Figure 4.
Additional modifications in the revised manuscript
- We have deleted opioid peptides released from M2 macrophages in Figure 2B, since it is not necessarily clear whether M1 or M2 macrophages secrete opioid peptides. Please find new Figure 2B, L174-L175 and L196 in the revised manuscript.
- We have added TGF-β into new Figure 2A, according to the description about the TGF-β-induced polarization of M0 to M2 macrophages in the text (L130-L132).

Reviewer 2 Report
This is an excellent review paper about neuroimmune interactions in peripheral nociception. The authors have done an excellent job in summarizing and their own work and work of others in the field. This a a very hot topic and article is well timed. Illustrations are also excellent as well and overall its was pleasure to read this manuscript.
I have not find any typos or correct statements.
Author Response
Comments
Responses to Reviewer #1
We thank the reviewer for the positive comments to our manuscript.
Additional modifications in the revised manuscript
- We have deleted opioid peptides released from M2 macrophages in Figure 2B, since it is not necessarily clear whether M1 or M2 macrophages secrete opioid peptides. Please find new Figure 2B, L174-L175 and L196 in the revised manuscript.
- We have added TGF-β into new Figure 2A, according to the description about the TGF-β-induced polarization of M0 to M2 macrophages in the text (L130-L132).
